# Optimal Dosing and Patient Selection for Electrochemotherapy in Solid Abdominal Organ and Bone Tumors

**DOI:** 10.3390/bioengineering10080975

**Published:** 2023-08-18

**Authors:** Cora H. Martin, Robert C. G. Martin

**Affiliations:** Division of Surgical Oncology, Department of Surgery, University of Louisville School of Medicine, Louisville, KY 40202, USA

**Keywords:** electroporation, electrochemotherapy, Bleomycin, solid organs, European Standard Operating Procedures for Electrochemotherapy (ESCOPE) guidelines, local disease control rate

## Abstract

The primary aim of this study was to analyze studies that use electrochemotherapy (ECT) in “deep-seated” tumors in solid organs (liver, kidney, bone metastasis, pancreas, and abdomen) and understand the similarities between patient selection, oncologic selection, and use of new procedures and technology across the organ systems to assess response rates. A literature search was conducted using the term “Electrochemotherapy” in the title field using publications from 2017 to 2023. After factoring in inclusion and exclusion criteria, 29 studies were analyzed and graded based on quality in full. The authors determined key patient and oncologic selection characteristics and ECT technology employed across organ systems that yielded overall responses, complete responses, and partial responses of the treated tumor. It was determined that key selection factors included: the ability to be administered bleomycin, life expectancy greater than three months, unrespectability of the lesion being treated, and a later stage, more advanced cancer. Regarding oncologic selection, all patient cohorts had received chemotherapy or surgery previously but had disease recurrence, making ECT the only option for further treatment. Lastly, in terms of the use of technology, the authors found that studies with better response rates used the ClinporatorTM and updated procedural guidelines by SOP. Thus, by considering patient, oncologic, and technology selection, ECT can be further improved in treating lesions in solid organs.

## 1. Introduction

Electrochemotherapy (ECT) continues to advance as more clinical trials produce improved results regarding the treatment of “deep-seated” solid-organ tumors in challenging locations. These advances come from the improvements in the technology used in ECT—electrode geometry, dosing, and needle variation—and understanding the optimal patient and oncologic selection for ECT. ECT, more specifically, uses reversible electroporation through high-voltage electrical pulses to temporarily increase the membrane porosity for improved diffusion of chemotherapeutics into the cell, leading to increased chemotherapeutic activity. In the last five years, due to the new SOP guidelines and further clinical studies, ECT has transitioned from solely treating cutaneous tumors using standard ECT to treating complicated tumors in challenging locations using variable ECT [1]. With the new age of medical technology and improved patient selection, ECT has begun to treat solid organ tumors in the chest and abdomen to improve tumor responses and necrosis of tumor tissue, leading to improved pain levels and quality of life [2,3,4]. This transition is significant because of the limited treatment options available after failure of first-line or second-line therapies in these malignancies.

A significant number of reviews regarding ECT have been published in recent years due to this transition period. Most of these publications discuss the advances within only a specific organ system, the relationship between tumor biology, or the procedural aspects of ECT [5,6,7,8,9]. There has not yet been a review that has summarized the optimal patient selection and use of ECT in solid organs. Therefore, this systematic review aims to identify the key patient cohort selection characteristics and procedural aspects of ECT that elicit favorable responses which create treatment success.

## 2. Methods

A comprehensive literature review was performed using PubMed and the Library of Congress. The initial search was conducted using the term “Electrochemotherapy” in the title field, using publications from 2017 to 2023. The initial search yielded 357 results. One author independently screened the titles and abstracts of all articles identified by the primary search strategy. Inclusion criteria were limited to English articles and included human observational and comparative cohort studies focused on solid organs—bone metastasis, kidney, liver, pancreas, or lung treatments. These reference lists were further examined to identify additional studies not captured by the primary literature search, and an additional two articles were hand-selected and added to the search. From the combined 359 articles, 330 were excluded after being screened for the following criteria: the publication was not a human study (n = 101), the publication was not in English (n = 2), the publication was a systematic review or meta-analysis (n = 32), and the publication was not focused on solid organs—bone metastasis, kidney, liver, pancreas, or lung treatments (n = 195). Based on the inclusion and exclusion criteria, the author evaluated the remaining 29 articles for patient selection, type of therapy, response rate, progression of free survival, and overall survival (Figure 1). Each of the 29 articles was graded using the MINORS, the methodological index for non-randomized studies, a scoring system that assigns a strength score based on experimental design and number of participants in the study [10]. The definition of electrochemotherapy follows The European Standard Operating Procedures for Electrochemotherapy (ESCOPE) guidelines. ESCOPE states that potential contraindications to electrochemotherapy include poor renal function (less than CrCl 50 mL/min), ventricular arrhythmias, disruption of pacemakers, pulmonary fibrosis, and previous lifetime exposure to bleomycin above a stated threshold dose—less than 400 IU/m^2^ units [10,11,12,13]. Enduring conflicts and questions were resolved following the senior author (RM) review.

The 29 selected papers were categorized into organ systems (Liver, Pancreas, Kidney, Bone Mets, and Stomach) in separate tables. Within these tables, the authors focused on the patient cohort, procedural technology used, and the main results. In the patient cohort section, the authors recorded: gender, age, type of cancer, size of tumor, and the number of tumors treated. In the procedural section, the authors recorded: the drug used, the administration method, the needle configuration, the pulses and use of the electric field, a treatment window, and whether the paper utilized the ESCOPE guidelines. The main results recorded were those that provided curative intent results—overall response, OR; complete response, CR; partial response, PR; stable disease, SD; overall survival, OS; and pain-free survival, PFS—and palliative approach results—improvement of pain or quality of life based on the scale used. From there, the authors used the MINORS grading system to grade the quality of the studies.

## 3. Results

### 3.1. Pancreas ECT

Five studies focused on treating tumors in patients (female and male) with stage 3 LAPC (locally advanced pancreatic adenocarcinoma) or stage four metastatic disease. The median age for the studies that reported age was reported as 64 years (range 62–68). The median number of tumors treated per study was 21 (range 5–25), and the median tumor length was 52 mm (range 33–68). Three studies used ECT for a curative intent while two studies used ECT for palliation of symptoms—nausea, bleeding, and pain [14]. All studies used a concurrent intravenous drug administration, with four studies utilizing bleomycin at 15,000 IU/m^2^ and one study, cisplatin at 1 mg per mL per 1 cm tumor. Two studies provided a treatment window which started pulses 8 to 28 min after bleomycin injection. The CliniporatorTM formulated the correct electroporation pulses, and a hexagonal formation was utilized during open surgery (n = 5). In terms of pulses and electric fields, all four studies used at least eight pulses for a duration of 100 μs with an electric field ranging from 400–1000 V/cm, as per the user manual; three studies explicitly stated that they utilized the ESCOPE guidelines.

In terms of patient selection, all studies gave inclusion and exclusion criteria consistent with the following. Inclusion criteria included: confirmed pancreatic adenocarcinoma; unresectable pancreatic cancer; age > 18 years old; failure of at least one line of chemotherapy; life expectancy of more than 3–6 months; ECOG performance status of 0 to 2; and that subjects have recurrence or treatment refractory disease. Exclusion criteria included: pregnant women, patients with a pacemaker that could not be deactivated, patients with severe cardiac arrhythmia (other than sinus rhythm), allergy to bleomycin, kidney disfunction, and a history of pulmonary edema or fibrosis.

In terms of oncologic selection, all patients in all studies received chemotherapy before the ECT procedure. Thus, ECT was administered due to either local recurrence after this therapy or due to a stable disease or partial response after chemotherapy.

Overall results included: median OS was 29.5 months with IRE and chemotherapy [15]; 24/25 patients had LDCR with an overall MS, median survival, 11.5 months [16]; 15/17 lesions had a complete response after ECT [17]; and according to the MR, magnetic resonance imaging parameters, there was a 9/11 PR, according to the Choi criteria there was a 18/18 PR, and according to the PERCIST there was a 6/10 PR in lesions treated [18]. The PERCIST criteria serve as a quantitative assessment that evaluates metabolic tumor rates after treatment; the RECIST criteria constitute an anatomic assessment that evaluates the response by comparing time interval change to tumor size; the Choi criteria consider the tumor’s response to treatment with at least a 10% reduction in size and 15% reduction in density [19,20] (Table 1).

### 3.2. Liver ECT

A total of 17 studies focused on treating patients (female and male) with hepatocellular carcinoma (n = 10), cholangiocarcinoma (n = 3), colorectal cancer (n = 4), and renal cell carcinoma with a liver metastasis (n = 1) with ECT. The median age for the studies was 64 years (range 47–75). The median number of tumors treated was 12 per study and the median length of each tumor was 24 mm (range 12–46 mm). All studies used an intravenous drug administration with a treatment window that started pulses 8 to 28 min after injection. 16 studies utilized bleomycin at 15,000 IU/m^2^ and 1 study used bleomycin at 29.4 mg. All studies provided information on electrodes: 8/17 studies utilized 16–20 cm long electrodes and 12/17 used electrodes having an active tip of 1.5–4 cm. A total of 13 studies used 8–96 pulses for a duration of 100 μs with an electric field ranging from 730–1000 V/cm at 1–5 kHz. Methods of electrode administration included open surgery (n = 10), percutaneous placement (n = 5), and laparoscopic administration (n = 1). Four studies explicitly stated they utilized ESCOPE guidelines, while five studies utilized SOP guidelines.

In terms of patient selection, all studies gave inclusion and exclusion criteria consistent with the following. Inclusion criteria included: unresectable primary or recurrent hepatic lesions, liver-dominant metastatic disease, good performance status, life expectancy of more than 3 months, patient age > 18, target tumor in proximity (<1 cm) to vital blood vessel(s) or challenging location, and a performance status greater than 70% according to the Karnofsky Performance Status of the World Health Organization. Exclusion criteria included: pregnancy, allergy to bleomycin, heart impairment, arrhythmias or pacemakers that could not be disrupted, impaired kidney function, and that prior cumulative exposure to bleomycin was limited to <400,000 IU/m^2^.

In terms of oncologic selection, all patients in all studies received chemotherapy, immunotherapy, or previous had liver resection before the ECT procedure. ECT was administered due to local recurrence after tumors failed to respond to the given therapy or due to a stable disease or partial response after chemotherapy or other local therapies.

ECT performed best in lesions between 3 and 6 cm with an OR of 85.7%, CR of 61.9%, and PR of 23.8%, with a mean PFS of 9 months and OS of 11.3 months. In sum, 90.5% of lesions treated in this study were in challenging locations: tumors located directly next to or in close vicinity (<1 cm) to vital veins, significant arteries, areas of the pancreas, and the heart [22]. Another study had similar response rates: 84.4% CR, 12.5% PR, and a median PFS of 12 months. Interestingly, this study also looked at differences in rates between different electrode variations, in which the patients treated with variable geometry’s CR was 81.8%, while use of a fixed hexagonal geometry had a CR of 85.7% [23]. On the other hand, most case studies (n = 5) did not provide any kind of response rate, but did provide confirmation of preservation of large blood vessels and necrosis of treated areas [24,25] (Table 2 and Appendix A).

### 3.3. Kidney ECT

Two studies focused on treating tumors in the kidney with ECT. The two case studies included utilized two females with stage 3 renal cancer or renal cell carcinoma. One female, 85, displayed two tumors, 25 × 16 mm and 43 × 34 mm, while the other female, 62, displayed a tumor measuring 15 × 11 mm. Both studies used ECT with a curative intent approach. Both studies used an intravenous drug administration and bleomycin at 15,000 IU/m^2^ and started pulses 8 min after bleomycin injection after electrodes were percutaneously placed. Neither study utilized the ESCOPE guidelines. Before ECT treatment, the 85-year-old woman had previously undergone chemotherapy, radiotherapy, and a proctectomy which resulted in a local recurrence. In the case of the 62-year-old woman, she had received a right radical nephrectomy and cryoablation before having a local recurrence which was to be treated with ECT. In the case study with the 61-year-old female, a 6-month CT found no evidence of residual disease, while in the case study of the 85-year-old female, a CT showed tumor necrosis and complete response of tumor after treatment [31,32] (Table 3).

### 3.4. Bone Metastases ECT

Three studies focused on treating metastatic bone tumors in males and females with ECT. The median age was found to be 63 (range of 59–66). The median number of treated tumors per study was 38 (range of 32–102). The median of tumors axially was 37 mm (range 3–140 mm), coronally was 52 mm (range 12–160 mm), and sagittally was 50 mm (range 12–160 mm). All studies used ECT as a curative intent approach. Two studies used an intravenous drug administration and bleomycin at 15,000 IU/m^2^ and started pulses 8 to 28 min after bleomycin injection. Two studies used 16–20 cm percutaneously placed needle electrodes with a 3–4 cm active tip for eight pulses for 100 μs duration with a local electric field of 1000 V/cm. Three studies explicitly stated they utilized ESCOPE guidelines.

In terms of patient selection, inclusion criteria included: age greater than 18, appendicular or axial skeleton by metastatic carcinoma or melanoma with impending fracture, indication for internal fixation with a nail, and local treatment of ECT. Exclusion criteria included: pregnancy, allergy to bleomycin, previous dose of >40,000 IU of bleomycin in lifetime, kidney dysfunction, metallic devices within field of electrical pulses, pulmonary fibrosis, and edema.

In terms of oncologic selection, all patients underwent at least first-line or more chemotherapy, hormonal therapy, or radiotherapy and then had persistent or local disease recurrence.

Pain control was assessed using the visual numeric scale, which is a self-reporting pain scale that ranges from 0 (no pain) to 10 (maximum pain) [33]. Before ECT, pain was a 5.1, at first follow-up it was 2.3, and at later follow-up it was 2; there was a decrease in pain in 23/29 patients [2]. Furthermore, bone recovery (ossification of osteolytic lesions) was observed in sixteen patients: CR in ten patients, PR in six patients, and SD in nine patients [2]. According to PERCIST criteria, 31.4% of patients saw an OR response and, according to PERCIST criteria, 40.4% of patients saw an OR response [3] (Table 4).

The overall analysis of these studies demonstrates that ECT is effective in treating deep-seated solid organ tumors and shows similarities across patient and oncologic selection criteria.

## 4. Discussion

Cancer treatments with ECT have proven to be a successful and minimally invasive treatment method for certain patients with advanced solid organ staged cancers. The standard ECT for cutaneous malignancies has been demonstrated thoroughly, but the use of ECT on solid organ tumors in the chest and abdomen is beginning to show encouraging results. This systematic review has demonstrated that ECT has been a treatment option for patients after failing first- and second-line therapies based on progression or intolerance to therapy. The diseases were also more commonly to be locally advanced staged diseases that were not amenable to surgical resection. This is the first review to outline and organize the use of ECT based on organ site, reported optimal dose of chemotherapy, and patient selection.

The single-largest cause of ECT use was in treating metastatic or locally advanced hepatic malignancies. They were all recurrent or locally advanced disease metastases with a close proximity (<1 cm) to major blood vessels or bile ducts [22]. ECT in lesions close to the heart (lung or liver) showed no effect on cardiac function or rhythm changes. Ten patients were monitored with Holter electrocardiographic signals during and after treatment of ECT, by which no significant changes were found [35].

Hepatocellular carcinoma tumors were found to have the best local tumor control rate (overall response rate >75% at 3 months) after ECT, followed by colorectal cancer [36]. Studies that yielded CR rates, ranged from 75–88% [22,23,28,29,30], had selected patients who had undergone chemotherapeutic treatment but subsequently had a recurrence or presented with SD, for whom ECT was presented as an alternative method of treatment for curative or palliative purposes. All patients, most importantly, needed to be able to be administered bleomycin for the procedure and have a >3-month life expectancy. Furthermore, all studies selected provided a conclusionary statement that ECT responded to the patients’ recurrent disease and was feasible, safe, and effective. Studies included, however, were mostly non-comparative pilot prospective studies in which ECT was only being studied in terms of safety and feasibility rather than compared to other treatments. MINORS grades for the studies ranged from 7 to 12, out of 16. This was largely due to patients not being included consecutively in the study, the study being biased in that only one institute was selecting patients, or an inability to provide power calculations of the study population. Thus, although the studies provided encouraging results for this treatment, more comparative studies across multiple institutions with more statistics are required before definitive conclusions can be drawn.

Pancreatectomy for stage 1 and 2 patients, although effective, has an adverse event expectation of 30–40%, which includes pancreatic fistulas, hemorrhages, pancreatitis, and delayed gastric emptying [37]. ECT has been used to obtain responses in later stage disease (stage 3—LAPC) to treat tumors and provide pain improvement. Like patients being treated with liver malignancies, it seems as though the combination of recent chemotherapeutic treatment increases the effects of the ECT treatment. Patients with a combination of chemotherapy and IRE were found to have an encouraging median OS survival (29.5 months) compared to IRE alone (16 months) [15]. Furthermore, according to the EQ-5D-5L pain scale, patients’ pain score after treatment went from 2, pain and discomfort, to 1, no post-operative problems [21]. MINORS grades, ranging from 8 to 9 out of 16, revealed similar limitations to those of patients being treated for liver malignancies (as listed above).

Metastatic bone disease affects 4.9 million individuals in the United States alone [38]; in this disease, osteolytic lesions continually weaken bone density, making patients susceptible to fracture and breakage [2]. Patient inclusion criteria for an ECT procedure included having an impending fracture as well as disease recurrence after first-line chemotherapeutic failure. After ECT, patients experienced bone recovery from these lesions, as well as ORs (range 22.7–48%), according to both PERCIST and RECIST [2,3,34]. MINORS grades, ranging from 7–9 out of 16, revealed similar limitations to those of patients being treated for liver malignancies (as listed above).

Finally, studies regarding kidney cancer with ECT provided, again, similar patient and oncologic selection criteria compared to the other solid organ systems. Although encouraging results, both studies were case reports. Thus, it is not reasonable to assume that their results are definitive conclusions. Furthermore, most studies included, indicated the use of ESCOPE and the updated SOP guidelines. Although ECT is continuing to become a more standardized procedure, it is still not recognized by ESMO, the European Society of Medical Oncology, or the NCCN, the National Comprehensive Cancer Network, as a treatment for deep-seated tumors, even in reserved cases [39].

Systematic guidelines for ECT have been developed in the last five years, with the publishing of the SOP guidelines preceding the ESCOPE guidelines in 2006. Recent advancements in the procedure include the development of technology for accurately placing electrodes to ensure entire tumor coverage and combinational strategies such as ECT with IRE or immunotherapy [9].

With all these various advancements, bleomycin (BLM), a cancer drug created in 1962 which seems outdated, is used as the drug of choice in 26 of the 29 studies included in this review [40,41]. Thus, BLM is still the drug of choice for ECT procedures for various reasons. The first is that ECT relies on creating a permeable membrane to increase the cytotoxicity of cancer drugs to cause necrosis of tumor tissue through an electrical field. Compared to cisplatin, in which ECT potentiates the cytotoxic effect by 12-fold, ECT potentiates the cytotoxic effect of bleomycin up to 5000-fold, making it a highly effective cytotoxic drug during this procedure [42]. Secondly, BLM has been shown to have an abscopal effect: a systemic immune response mediated by the effects of this drug on the immune system [43]. This effect has been demonstrated in cutaneous and skin metastases, in which there is evidence that ECT with BLM, in combination with immunotherapies, can produce this effect [44]. In terms of dosing, all studies that used BLM used a dosage of 15,000 IU/m^2^ administered intravenously, which has continued to be the standard operating procedure for this treatment and the optimal dose for electrochemotherapy treatment, as it is the lowest effective dose providing the least amount of chemotherapeutic side effects, according to ESCOPE and SOP guidelines [10,11,12,13].

Another injection that gives this electro-chemo-effect is cisplatin, CDDP. New advances in categorizing the biology of specific tumors have shown that CDDP is better for use in ECT, compared to BLM, in specific instances and vice versa. More specifically, tumors subject to viral infection change their biology in such a way that HPV+ (human papillomavirus) and HPV- tumors responded comparably to ECT with BLM but were more sensitive to ECT with CDDP in HPV+ tumors [45]. On the other hand, radioresistant and naïve tumors showed an equal response to ECT with BLM, but radioresistant tumors were more resistant to ECT with CDDP [46] Thus, in different situations, either drug can be more favorable depending on tumor biology, electro-chemosensitivity, elicitation of an immune response, and physician preference.

Finally, calcium electroporation has been of interest when used as the drug of choice for ECT. This treatment utilizes supraphysiological doses of calcium internalized by electroporation and causes cell death. The advantages of this treatment include that it is non-mutagenic and has long durability [47,48]. In this review, calcium electroporation was only found to have been used once in the articles selected and, in combination with IRE, has a median OS of 19 months compared to a chemotherapy treatment, BLM, with IRE having a median OS of 29.5 months [15].

The pulses, electric field, and electrode placement are the other pieces of ECT which create a variety of treatments for each lesion. New programs such as the Cliniporator VITAE have provided physicians with a way to deliver both the optimal amount of electroporation pulses synchronized with electrocardiogram and the optimal voltage for the electrodes connected to the device. Before the delivery of pulses to the tumor, a sequence of pre-pulses allows the Cliniporator to assess if electrode pairs are delivering a current that is too low or too high, and it continues to monitor voltage throughout the procedure [49]. Before treatment, infrared scans, computed tomography (CT), and 3D models based on MRI images are used to create an electrode geometry (electrochemotherapy). In the studies used, due to the smaller size of lesions, a hexagonal formation was used, and CT in several studies confirmed the coverage of the lesions in their entirety [16,23,27,29]. Thus, the use of the Cliniporator VITAE or a similar pre-treatment confirmation is critical for the successful treatment of the entire tumor.

In terms of the number of pulses and the strength of the electric field, the measurements of these aspects depend on the threshold between reversible and irreversible electroporation. Reversible electroporation occurs with the increase in membrane permeability, but the cell can regain homeostasis where irreversible electroporation bombards the cell with a high electric field and a high number of pulses, which induces cell death. Thus, the optimal pulse and electric field has been found to be eight pulses of 100 μs, with an amplitude of 100–1000 V, whereas IRE utilizes 80 to 100 pulses with an amplitude up to 3000 V [50]. Twenty-two out of the twenty-nine studies included used this range of electric field for this time duration and number of pulses. Thus, it is clear with the success of these studies that optimal pulses range from 8 to 28 with a voltage of 100–1000 V for a treatment time of up to 28 min in accordance with the SOP and ESCOPE guidelines. This provides the patient with a short treatment time and a voltage that is reversible rather than irreversible, so that non-malignant cells are preserved; most importantly, this leads to enhanced drug delivery to malignant cells in the treated region.

ECT has not been established as a first-line therapy for the solid organ tumors reviewed. All patients treated in this review had undergone first-line therapy—including chemotherapy, hormone therapy, surgery, or radiation—and experienced recurrence; thus, ECT was presented to them as a second-line or later therapy for their treatment. ECT outcomes were best in patients when the treatment was combined with a chemotherapy rather than calcium or IRE alone [15,27,29]. Thus, from the articles selected, ECT with these chemotherapies creates the best response for patients with solid organ tumors.

Finally, the last advancement in deep-seated tumor treatment with ECT is a combination strategy using other techniques in addition to ECT. ECT alone induces a local immune response and has been considered by several reviews to be a type of in situ vaccination [51,52]. However, in addition to the ECT procedure, combining immunotherapeutic drugs helps to enhance this local and systemic immune response after ECT treatment. Certain clinical studies have shown that ECT, in combination with immune checkpoint inhibitors such as anti-CTLA-4, anti-PD-1, and anti-PD-L1, has been shown to have local immune response effects. Another approach that has been studied is the combination of ECT with toll-like receptors like TLR3 and Flt3l, which help to recruit antigen-loaded and, activated intratumorally, cross-presenting dendritic cells [53,54,55,56,57].

ECT with irreversible electroporation, IRE, combines both reversible and irreversible electroporation to treat a tumor lesion. In irreversible electroporation, a high-voltage electrical pulse exceeds a specific threshold, overwhelming the cell and causing irreversible injury and death [8]. The combination of the two techniques, now deemed IRECT, treats tumors by creating an IRE zone that covers the entirety of the tumor and an RE zone that covers the area surrounding the tumor. This method not only allows for the complete apoptosis of cells within the tumor, but the RE zone also allows for the apoptosis of residual tumor cells while not permanently damaging the healthy cells surrounding the tumor [58]. Thus, several combinational techniques can be applied alongside ECT, making an ever-more-enhanced response dependent on tumor biochemistry and disease type.

There remain a few limitations for this review. This review is not inclusive of all types of ECT used, and specifically excludes soft tissue and surface-based malignancies (i.e, breast and melanoma). We focused on just the recent cases (within last 5–7 years) of “deep seated” solid organs based on the current clinically unmet need and recent expansion of these organs for ECT treatment. Limitations for ECT still remain overall, first demonstrated by the use of only BLM or cisplatin, regardless of the tumor type or tumor histology. It is critical for ECT to expand and evolve into more clinically relevant chemotherapies and begin to augment/ mirror systemic use (i.e., FOLFOX for metastatic colorectal, gemcitabine for pancreatic adenocarcinoma, etc.). The second limitation is in the transient but real immune effects that reversible and irreversible electroporation induce, and in expanding the combination of reversible and irreversible techniques with immuno-oncology therapies, especially in “cold solid organ” tumors. The last consideration is to optimize the dose of ECT used based on tumor volumes, patient volumes, and maximal tolerated doses for specific organs, which would further expand the knowledge and use of ECT in oncology patients.

## 5. Conclusions

Analysis of these studies shows that patient and oncologic selection plays an important role in the success of ECT in solid organ tumors in challenging locations. Regarding patient selection, characteristics similar across all systems analyzed included: the ability to be administered bleomycin, life expectancy greater than three months, unrespectability of the lesion being treated, and a later stage, more advanced cancer. Regarding oncologic selection, strikingly, across all the solid organs, all patient cohorts received chemotherapy or surgery previously but had disease recurrence, making ECT the only option for further treatment. The success of ECT not only came from the two kinds of selections but from advances in technology used in the ECT procedure, including using the ClinporatorTM and updated procedural guidelines by SOP, which are used to improve treatment in all solid organs. Thus, patient and oncologic selection criteria, according to the response rates of the clinical studies analyzed, can improve the success of ECT.

## Figures and Tables

**Figure 1 bioengineering-10-00975-f001:**
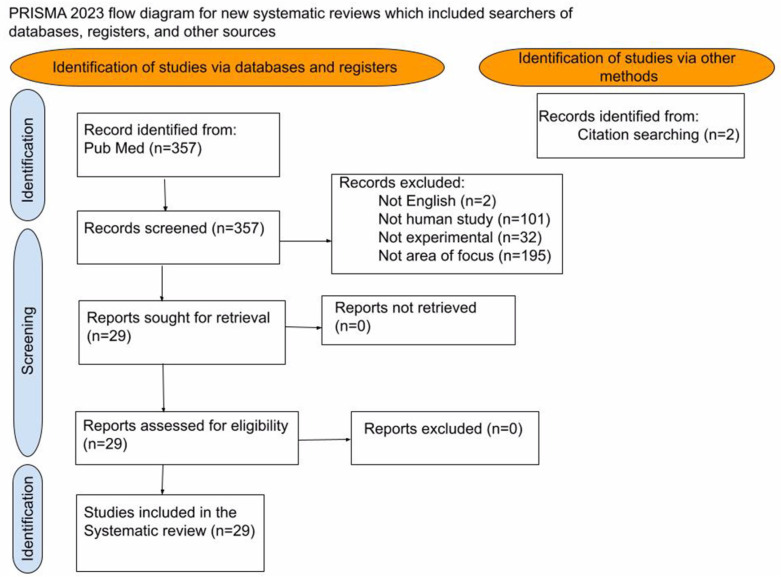
Prisma Diagram: Search methods for studies included.

**Table 1 bioengineering-10-00975-t001:** Electrochemotherapy clinical data in the pancreas.

Author, Year	# Pts	Main Test or Hypothesis	Patient Cohort	Procedural	Main Results	Grade
Rudno-Rudzinska et al. [15] 2021	13	IRE, ECT, and calcium electroporation techniques investigated as new strategies for treating pancreatic cancer	**Gender**: male (11), female (12) **Age**: median (62) range (38–76) **Type of cancer**: PAC: stage two to four **Size of tumor**: median: 33 mm (range 21–32 mm) **Number of tumors treated in study**: 24 **Curative or palliative**: curative	**Drug use**: Cisplatin-Ebwe 1 mg/1 mL, CaCl2 9 mg/mL 10% solution **Administration method**: Intravenous, open surgery **ESCOPE**: YES **CliniporatorTM**: Not used	**Median OS**: 16 (IRE), 29.5 (IRE and CTH), 19 months (IRE and CaCl2), 10 months (CTH).	14/24
Izzo, Francesco [16] 2021	25	Assess local disease control rates and overall survival in locally advanced pancreatic cancer treated with electrochemotherapy	**Gender**: male (11), female (14) **Age**: median (68), range (48–68) **Type of cancer**: LAPC, stage 3 **Size of tumor**: median (51 mm), range (28–99 mm) **Number of tumors treated in study**: 25 **Curative or palliative**: curative	**Drug use**: Bleomycin, 15,000 IU/m^2^ **Administration method**: Intravenous, laparotomy **Needle configuration**: Hexagonal, linear formation **Number needle**: 8, 7 **Length of needle**: 16–24 cm, active tip 3–4 cm **Pulses**: 8–96 pulses at 400–1000 V/cm of 100 μs duration at 1–5000 Hz **ESCOPE**: YES **CliniporatorTM**: Used	**One month after ECT**: 76% 19/25 patients were in PR and 5/25 20% in SD: LDCR 24/25 96%. **Six months after ECT**: 11/25 44% in PR, 3/25 12% in SD: 14/25 LDCR 56%. **Overall MS 11.5 months**: Patients treated with fixed geometry: 6 months, variable geometry: 12 months.	9/16
Casadei, Riccadro [21] 2020	5	Assess whether intraoperative ECT could be proposed as additional therapy in treating LAPC	**Gender**: female (3), male (2) **Age**: median (62), range (56–68) **Type of cancer**: LAPC, stage 3 **Size of tumor**: median: 32 mm (range: 22–43 mm) **Number of tumors treated in study**: 5 **Curative or palliative**: palliative	**Drug use**: Bleomycin, 15,000 IU/m^2^ **Administration method**: Intravenous, open surgery **Needle configuration**: Oval, circle formation **Number needle**: Range 4–6 **Length of needle**: 16–20 cm, active tip 3–4 cm **Pulses**: 8 electric pulses, 100 μs duration, 1000 V/cm generated **Treatment window**: 8–40 min after end of bleomycin injection **ESCOPE**: NO **CliniporatorTM**: Used	**The EQ-5D-5L scale**: Case 1, Case 2, Case 3 pain/discomfort score went from 2 (slight pain/discomfort) to 1 post-operative no problems.	8/16
Granata Vincenza [17] 2019	21	Evaluate diagnostic performance of diffusion weighted imaging and by diffusion kurtosis imaging to assess pancreatic tumors treated with ECT	**Gender**: (11) female, (10) male **Age**: N/A **Type of cancer**: LAPC, stage 3 **Size of tumor**: 52 mm (22–99 mm) **Number of tumors treated in study**: 21 **Curative or palliative**: curative	**Drug use**: Bleomycin, 15,000 IU/m^2^ **Administration method**: Intravenous, open surgery **Needle configuration**: Linear or hexagonal **Pulses**: 8–96 pulses of 400–730 V and 1000 V/cm, of 100 μs duration, at 5000 Hz repetition frequency **ESCOPE**: NO **CliniporatorTM**: Used	**CR**: Average of 20.5 months ECT 15/17 regions **SD**: 3 patients	9/16
Granata, Vicenza [18] 2017	19	Report early imaging assessment of ablated area post-ECT in patients with LAPC	**Gender**: (9) female, (10) male **Age**: median (67), range (48–80) **Type of cancer**: LAPC **Size of tumor**: median: 51 mm (range 22–99 mm) **Number of tumors treated in study**: 19 **Curative or palliative**: palliative	**Drug use**: Bleomycin, 15,000 IU/m^2^ **Administration method**: Intravenous, laparotomy **Needle configuration**: Linear, hexagonal **Pulses**: 8–96 pulses of 400–1000 V and 910–1000 V/cm of 100 μs duration at 1–5000 Hz **Treatment window**: 8–28 min after bleomycin injected **ESCOPE**: YES **CliniporatorTM**: Used	**MR parameters**: 9/11 PR, 2/11 SD **Choi criteria**: 18/18 PR **PERCIST criteria**: 6/10 PR, 3/10 SD, 1/10 PD	8/16

OS = overall survival, MR = magnetic resonance image parameters, IRE = irreversible electroporation, CTH = control chemotherapy group, PR = partial response, SD = stable disease, LDCR = local disease control rate, MS = median survival, LAPC = locally advanced pancreatic cancer, CR = complete response.

**Table 2 bioengineering-10-00975-t002:** Electrochemotherapy clinical data in the liver.

Author, Year	# Pts	Main Test or Hypothesis	Patient Cohort	Procedural	Main Results	Grade
Iezzi, Roberto et al. [26] 2023	5	Test feasibility and safety of ECT for treatment of complex liver tumors unit for thermal ablation	**Gender**: (3) male, (2) female **Age**: median (67), range (56–81) **Type of cancer**: primary and recurrent HCC **Size of tumor**: range (26–54 mm) **Number of tumors treated in study**: 42 **Curative or palliative**: palliative	**Drug use**: Bleomycin, 15,000 IU/m^2^ **Administration method**: Intravenous, percutaneous **Needle configuration**: Pentagon **Number needles**: Max 6 **Length of needle**: 16 cm with 4 cm active part, 20 cm with 3 cm active part **Treatment window**: 8 to 40 min after bleomycin injection **ESCOPE**: NO **CliniporatorTM**: Used	**Complete lesion coverage**: Post-procedural cone beam CT **One month CT**: OR 100% (four CR and one PR)	11/16
Edhemovic, Ibrahim [27] 2020	39	Evaluate long-term effectiveness and safety of ECT in treatment of unresectable CRC metastases	**Gender**: male (28), female (11) **Age**: median (63), range (35–81) **Type of cancer**: CRC stage 4 **Size of tumor**: range (0.3–6 cm) **Number of tumors treated in study**: 84 **Curative or palliative**: curative	**Drug use**: Bleomycin, 15,000 IU/m^2^ **Administration method**: Intravenous, open surgery **Needle configuration**: Hexagonal **Number needles**: 6 **Pulses**: Trains of eight pulses, pulse 100 μs long **Treatment window**: 8 min after bleomycin injection **ESCOPE**: SOP procedure **CliniporatorTM**: Used	**mRECIST criteria**: OR was 75% (63% CR, 12% PR) 2% SD, 23% PD **Response per patient**: 44%CR, 15% PR, 2.5% SD, and 38.5% PD **Average diameter of treated metastases 2 cm**: Response of smaller metastases, up to 3 cm, significantly better than large metastases (larger than 3 cm)	10/16
Boc, Nina [28] 2018	2	Aim of the study was to characterize ultrasonographic findings during and afterECT of liver tumors to determine ablation zone and verify coverage of treated tumor	**Gender**: male (1), female (1) **Age**: 56, 59 Type of cancer: CRC **Size of tumor**: 25 mm and 16 mm **Number of tumors treated in study**: 2 **Curative or palliative**: curative	**Drug use**: Bleomycin, 15,000 IU/m^2^ **Administration method**: Intravenous, open surgery **Needle configuration**: Rectangular pattern **Number needles**: 24 **Length of needle**: 3 cm long individual needle electrodes **Pulses**: 8 pulses of 100 μs duration, 400 V/cm and limit maximum current delivered to the tissue below 50 A **Treatment window**: 8 min after bleomycin injection **ESCOPE**: SOP procedure **CliniporatorTM**: Used	**Patient 1**: After 7 months: CR, tumor size reduces from 20 mm to 11 mm **Patient 2**: After 3 months 19 to 13 mm in diameter	10/16
Djokic Mihajlo [29] 2018	10	Aim of testing safety and effectiveness of ECT on CRC metastasis	**Gender**: female (2), male (7) **Age**: median (69), range (57–78) **Type of cancer**: HCC **Size of tumor**: median (24) mm, range (8–41 mm) **Number of tumors treated in study**: 17 **Curative or palliative**: curative	**Drug use**: Bleomycin, 15,000 IU/m^2^ **Administration method**: Intravenous, open surgery **Needle configuration**: Hexagonal geometry **Number needles**: 4–7 **Length of needle**: 20 cm needle electrodes with 3 or 4 cm active part **Pulses**: Trains of 8–24 electric pulses, 100 μs delivered **Treatment window**: 8–28 min after bleomycin injection **ESCOPE**: SOP **CliniporatorTM**: Not used	**CR**: 3–6 months was 80% per patient and 88% per treated lesion, after 20.5 months 15/17 lesions **CR** was obtained	12/16
Gasljevic, Gorana [30] 2017	7	Evaluate effectiveness and feasibility of ECT in patients with CRC metastases	**Age**: N/A **Type of cancer**: CRC **Size of tumor**: N/A **Number of tumors treated in study**: 13 **Curative or palliative**: curative	**Drug use**: Bleomycin, 15,000 IU/m^2^ **Administration method**: Intravenous, open surgery **Number needles**: 2 **Length of needle**: 1.2 mm diameter with 4 cm non isolated tip length **ESCOPE**: NO **CliniporatorTM**: Used	ECT induced coagulation neurosis in treated area of tumor and a narrow band of normal tissue: 85% **CR** Preserved functionality of vessels larger than 5 mm in diameter	10/16

OS = overall survival, MR = magnetic resonance image parameters, IRE = irreversible electroporation, PR = partial response, SD = stable disease, PD = progressive disease, LDCR = local disease control rate, MS = median survival, CR = complete response, CT = computer tomography, LTC = long-term care, HCC = hepatocellular carcinoma, CRC = colorectal cancer, PFS = pain-free survival, IC = incomplete response, CCA = cholangiocellular carcinoma.

**Table 3 bioengineering-10-00975-t003:** Electrochemotherapy clinical data in the kidney.

Author, Year	# Pts	Main Test or Hypothesis	Patient Cohort	Procedural	Main Results	Grade
Mastrandrea, Giovanni [31] 2021	1	Case study of 85-year-old woman with localized kidney cancer using ECT treatment	**Gender**: female **Age**: 85 (case study) **Type of cancer**: kidney stage 3 **Size of tumor**: 25 mm × 16 mm, 43 mm × 34 mm **Number of tumors treated in study**: 2 **Curative or palliative**: curative	**Drug use**: Bleomycin, 15,000 IU/m^2^ **Administration method**: Intravenous, percutaneous **Number**: 4 **Treatment window**: 8 min after bleomycin injection **ESCOPE**: NO **CliniporatorTM**: Used	No procedure-related complications; shorter periods of hospitalization and convalescence were needed with respect to standard surgery; ceCT showed complete tumor necrosis without residual viable tumor tissue in treated area; Obtained a **CR** without renal functional or quality of life impairment.	5/16
Andresciani, F [32] 2020	1	ECT used to treat 61-year-old female with renal cell carcinoma	**Gender**: female **Age**: 61 **Type of cancer**: kidney cancer **Size of tumor**: 15 mm × 11 mm **Number of tumors treated in study**: 3 **Curative or palliative**: curative	**Drug use**: Bleomycin, 15,000 IU/m^2^ **Administration method**: Intravenous, percutaneous **Length of needle**: (0.5–2.5 cm) distance between electrodes **Number needles**: 4 **Pulses**: 80 voltage 500–3000 V/cm, maximum deliverable current 50 A **Treatment window**: 8 min after bleomycin injection **ESCOPE**: NO **CliniporatorTM**: Used	At 6-month CT no evidence of residual disease was found.	4/16

OS = overall survival, PR = partial response, SD = stable disease, PD = progressive disease, CR = complete response, CT = computer tomography.

**Table 4 bioengineering-10-00975-t004:** Electrochemotherapy clinical data in bone metastases.

Author, Year	# Pts	Main Test or Hypothesis	Patient Cohort	Procedural	Main Results	Grade
Cevolani, Luca et al. [2] 2023	32	Test reduction in pain and local or systemic complication after undergoing ECT	**Gender**: (15) male, (17) female **Age**: median (66), range (38–88) **Type of cancer**: metastatic bone disease **Size of tumor**: axial length: 35 mm (17–55 mm), coronal: 63 mm (12–160 mm), sagittal: 63 mm (12–160 mm) **Number of tumors treated in study**: 32 **Curative or palliative**: curative	**Drug use**: Bleomycin, 15,000 IU/m^2^ **Administration method**: Intravenous, percutaneous **Length of needle**: 16–20 cm with an active tip of 3–4 cm **Pulses**: 8 pulses of 1000 V/cm with local electric field of 350 V/cm **Treatment window**: 8–30 min after bleomycin injection with an intramedullary nail implanted after treatment **ESCOPE**: YES **CliniporatorTM**: Used	**RECIST**—CR 3%, PR 45%, SD 45%, PD 7%; **PERCIST**—CR 7%, PR 21%, SD 17%, PD 55%; **Pain control**: Before ECT was 5.1 +/− 3, at first follow-up was 2.3 +/− 3.2, later follow-up 2 +/− 2.8 versus baseline—decrease in pain observed in 23/29 patients; **Bone response**: Bone recovery observed in 16 patients: **CR** 10 patients, **PR** 6 patients, **SD** 9 patients, **LDP** 4 patients.	7/16
Campanacci, Laura [34] 2022	38	Whether ECT is a feasible and effective treatment for metastatic bone disease	**Gender**: (13) male, (25) female **Age**: median (59), range (41–91) **Type of cancer**: bone metastases **Size of tumor**: axial: 49 mm (13–120 mm), coronal: 52 mm (16–120 mm) sagittal: 50 mm (16–120 mm) **Number of tumors treated in study**: 38 **Curative or palliative**: curative	**Drug use**: Bleomycin, 15,000 IU/m^2^ **Administration method**: Intravenous, percutaneous **Length of needle**: 12–20 cm length, 3–4 cm active part **Number needles**: 3–11 **Pulses**: 8 pulses of 1000 V/cm, 100 μs duration **Treatment window**: 8–30 min after bleomycin injection **ESCOPE**: YES **CliniporatorTM**: Used	**PERCIST**: 14% CR, 22% PR, 14% SD, 50% PD; **RECIST**: CR 9%, PR 16%, SD 59%, PD 16%; Pain: Overall 68% saw pain reduction; **Before ECT**: no: 11%, mild: 8%, moderate: 47%, severe: 34%; **Early FU**: no: 27%, mild: 43%, moderate: 20%, severe: 10%; **Late FU**: no: 47%, mild: 33%, moderate: 7%, severe: 13%; Among the 68%: 7 (2 CR and 5 PR) saw **bone recovery**, 17 w/o change + 1 with DP.	9/16
Campanacci, Laura [3] 2021	102	Confirm safety and efficacy of ECT in patients with bone metastases	**Gender**: (42) males, (60) females **Age**: median (63), range (38–91) **Type of cancer**: bone metastases **Size of tumor**: axial 37 mm, 3–140 mm), coronal (44 mm, 3–120 mm), sagittal (46 mm, 3–120 mm) **Number of tumors treated in study**: 102 **Curative or palliative**: curative	**Drug use**: Bleomycin, 15,000 IU/m^2^ **Administration method**: Intravenous, percutaneous **Length of needle**: 16–20 cm needle electrode with 3 cm or 4 cm active tip **Number needles**: 2–11 **ESCOPE**: YES **CliniporatorTM**: Used	**Overall treatment**: **PERCIST criteria**: 31.4% OR (5.6% CR and 25.8% PR) 51.7% SD, 16.9% PD; **RECIST criteria**: 40.4% OR, (2.2% CR, 38.2 PR) 50.6% SD, 9% PD; **Patients with nail**: **RECIST criteria**: 45.5% PR, 50.0% SD, 4.5% PD; **PERCIST criteria**: 22.7% PR, 54.5% SD, 22.7% PD; **Patients w/o nail**: **RECIST criteria**: 3% CR, 35.8% PR, 50.7% SD, 10.5% PD; **PERCIST criteria**: 7.5% CR, 26.9% PR, 50.7% SD, 14.9% PD.	9/16

OS = overall survival, MR = magnetic resonance image parameters, PR = partial response, SD = stable disease, PD = progressive disease, LDCR = local disease control rate, MS = median survival, CR = complete response, CT = computer tomography, LTC = long-term care, PFS = pain-free survival, FU = follow-up, local disease progression = LDP.

## Data Availability

Data available on request to the corresponding authors.

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
