# Peer review of "Optimal Dosing and Patient Selection for Electrochemotherapy in Solid Abdominal Organ and Bone Tumors"

_bioengineering, 2023, doi:10.3390/bioengineering10080975_

Round 1

Reviewer 1 Report

The paper is interesting and well written. It is acceptable for publication in the current form

Minor english editing

Author Response

Thank you for your comments and review.

Reviewer 2 Report

The systemic review article by Cora Martin and Robert CG Martin focuses on the patient selection, oncologic selection and technology selection of ECT on solid organ tumors. There have been numerous reviews on ECT, however the reviewer agrees that “There has yet to be a review that has summarized the optimal patient selection and use of ECT in solid organs.” The reviewer appreciates the value of this review “to identify the key patient cohort selection characteristics and procedural aspects of ECT that elicit favorable responses that create treatment success.”

The reviewer greatly acknowledges the efforts in the literature review. The contents are well organized and good summaries were made. A few major concerns and minor questions/suggestions listed below.

1. Tables are too long.

For example, Table 2. Electrochemotherapy Clinical Data in the Liver. has almost 8 pages. According to the publisher guidelines, “Very large tables, or many different tables showing similar cases, may be included in an Appendix or as supplementary data.”

https://www.mdpi.com/authors/layout#_bookmark40

2. Better summary of the findings are recommended:

Since the main focus of this article is on the on the patient selection, oncologic selection and technology selection of ECT on solid organ tumors, the reviewer would like to recommend tables summarizing the following 4 aspects for each of the tumor type:

·        Patient selection

·        Oncology selection

·        Technology selection: drug & administration

·        Technology selection: device & electrical pulses

3. “Optimal Dosing”, as in the title, remains unclear.

Questions remain, for example:

What should be the optimal choice of chemotherapy drug (BLM, Cisplatin and others) and their dose and timing?

What should be the optimal electroporation protocols?

What better approaches (both chemotherapy and electroporation) can be proposed to yield better cancer control and reduce side effects.

Other minor concerns, suggestions and questions:

1. The affiliation of the first author, which institute?

University of Louisville School of Medicine

@georgetown.edu

2. For systematic reviews, the keywords used, and the related literatures used for citation searching may need to be clarified.

3. Publications were only selected from 2017 to 2023. Clinical Electrochemotherapy can date back to the early 2000s.

4. In summarizing the clinical reports, information including prior treatment before ECT, whether or not the outcomes (e.g., OS, MS) are attributed to ECT alone or with other concurrent treatment may be relevant to be included in the tables.

5. What conditions (patient and oncologic selections) should ECT be prescribed, compared to alternative locoregional treatment methods such as IRE, since the authors mentioned IRE in the discussion?

6. In the abstract, “understand the similarities between patient selection, oncologic selection”. More discussion on the “similarities” across different tumor types is encouraged.

7. An acknowledgement section is encouraged.

Author Response

  1. Tables are too long.

For example, Table 2. Electrochemotherapy Clinical Data in the Liver. has almost 8 pages. According to the publisher guidelines, “Very large tables, or many different tables showing similar cases, may be included in an Appendix or as supplementary data.”

https://www.mdpi.com/authors/layout#_bookmark40   *** Thank you, all tables in the manuscript have a maximum of 5 studies and an appendix for the liver studies has been added.  ***

  1. Better summary of the findings are recommended:

Since the main focus of this article is on the on the patient selection, oncologic selection and technology selection of ECT on solid organ tumors, the reviewer would like to recommend tables summarizing the following 4 aspects for each of the tumor type:

  • Patient selection
  • Oncology selection
  • Technology selection: drug & administration
  • Technology selection: device & electrical pulses *** we have written this format requested into the results section.  Adding more tables would then contradict another Reviewers critiques that we have too many tables……  Thus we have left this format as presented. ***

  1. “Optimal Dosing”, as in the title, remains unclear.

Questions remain, for example:

What should be the optimal choice of chemotherapy drug (BLM, Cisplatin and others) and their dose and timing?   *** Yes this has been added. ***

What should be the optimal electroporation protocols?  *** This has been discussed and should currently follow the ESCOPE guidelines. ***

What better approaches (both chemotherapy and electroporation) can be proposed to yield better cancer control and reduce side effects.  *** This has been added to the discussion for optimal next steps in ECT. ***

  1. The affiliation of the first author, which institute?

University of Louisville School of Medicine @georgetown.edu  *** the first author is completing a research fellowship this summer at the University of Louisville, but is an undergraduate at Georgetown University, so they answer is really both. We have clarified this in the title page.  ***

  1. For systematic reviews, the keywords used, and the related literatures used for citation searching may need to be clarified. *** Thank you, clarified in the methods. ***

  1. Publications were only selected from 2017 to 2023. Clinical Electrochemotherapy can date back to the early 2000s. *** yes the first descriptions for ECT is back in the 2000’s but in our methods as described we focused on just the most recent current clinical data for this review to be relevant. ***

  1. In summarizing the clinical reports, information including prior treatment before ECT, whether or not the outcomes (e.g., OS, MS) are attributed to ECT alone or with other concurrent treatment may be relevant to be included in the tables. *** yes this has been further clarifies and emphasize to demonstrate that ECT was used after failure of 1st line and even later lines of treatment. ***

  1. What conditions (patient and oncologic selections) should ECT be prescribed, compared to alternative locoregional treatment methods such as IRE, since the authors mentioned IRE in the discussion? *** We have further added the current selection for ECT and recommendations for future studies and use. ***

  1. In the abstract, “understand the similarities between patient selection, oncologic selection”. More discussion on the “similarities” across different tumor types is encouraged. *** currently we did not find unique similarities and this has been edited in the abstract. ***

  1. An acknowledgement section is encouraged.  *** There is none. ***

Reviewer 3 Report

In this review, Robert C. G. Martin et. al., describes key patient and oncologic selection characteristics and ECT technology across organ systems that yielded overall responses, complete responses, and partial responses of the treated tumor. The studies were largely done in a systematic way. This review can be published after minor revisions.

1.     The keyword should be separated by a semi-column (;).

2.     In the scientific world the unit should be italic, authors are suggested to keep all the units in italic form.

3.     Figure 1 is not looking attractive authors are suggested to make it brighter and more attractive for the readers.

4.     Introduction is very small, the authors are suggested to elaborate the introduction with the more recent outcomes and what you are delivering in the review.

5.     There are a lot of spacing issues in the review article. The author is suggested to resolve this issue.

6.     There is no consistency in the references. Author are suggested to make a consistency in the reference section.

7.     Conclusion could be elaborate to some extent.

8.     There are many words that have been used direct without giving their full form. Authors are suggested to give the full form of the particular word during the 1st writing in the review.

9.     Authors are suggested to make 1-2 figures in the review article to make it attractive.

Author Response

In Response to Reviewer #3

This review can be published after minor revisions.  *** These have been edited. ***

  1. The keyword should be separated by a semi-column (;). *** corrected ***
  2. In the scientific world the unit should be italic, authors are suggested to keep all the units in italic form.
  3. Figure 1 is not looking attractive authors are suggested to make it brighter and more attractive for the readers. *** This is a standard PRISMA diagram.  I do not know how to make these figure more attractive. ***
  4. Introduction is very small, the authors are suggested to elaborate the introduction with the more recent outcomes and what you are delivering in the review. 
  5. There are a lot of spacing issues in the review article. The author is suggested to resolve this issue.
  6. There is no consistency in the references. Author are suggested to make a consistency in the reference section. *** corrected ***
  7. Conclusion could be elaborate to some extent. *** added ***
  8. There are many words that have been used direct without giving their full form. Authors are suggested to give the full form of the particular word during the 1stwriting in the review.

  1. Authors are suggested to make 1-2 figures in the review article to make it attractive. *** we believe the graphical abstract solves this critique. ***